photochemistry/materials science

titanium sheet, anodization, photoelectrical properties, photocurrent

**Authors for correspondence:**
Xiangping Chen
e-mail: 376254686@qq.com
Lishi Wang
e-mail: wanglsh@scut.edu.cn

†The authors contributed equally to this work.

This article has been edited by the Royal Society of Chemistry, including the commissioning, peer review process and editorial aspects up to the point of acceptance.

# Study on the photoelectrical performance of anodized titanium sheets

Xiangping Chen[1,†], Xin Li[2,†], Dedong Chen[1] and Lishi Wang[2]

[1]Jewelry Institute, Guangzhou Panyu Polytechnic, Guangzhou 511483, People's Republic of China
[2]School of Chemistry and Chemical Engineering, South China University of Technology, Guangzhou 510641, People's Republic of China

XC, 0000-0003-4426-8642

Anodization is a widely used method to obtain multicoloured oxidized titanium sheets. However, most researchers paid great attention to the colour-related properties instead of photoelectrical properties of titanium oxide film obtained by anodization. In this work, to study their photoelectrical properties, a series of multicoloured oxidized titanium sheets were prepared by anodization method, and UV–vis absorption and photocurrents were tested. The relationship between anodization voltages/anodization durations and photocurrents of titanium sheets was studied. Results show that titanium sheets have excellent photoelectrical performance. With the increase of anodization voltage, the number of UV–vis absorption peaks increased under visible light which means increasing absorption. When anodization duration increased, absorption band edge also increased in the visible light region, which means the band gap needed to produce charge transfer transition decreased. Under simulated sunlight and applied voltage of +0.4 V, photocurrent increased with the increase of either anodization voltage or anodization duration, and can be expressed by linear equations. In addition, anodization currents were recorded during anodization. Morphology, crystal structure and photoelectrical properties of anodized titanium sheets were characterized. The anodized titanium sheets can not only be used as decorative material in jewellery and architecture fields etc. but also are supposed to be used as photoelectrical catalyst in further work.

## 1. Introduction

Titanium has high corrosion resistance and chemical stability, and is frequently used in implanting field [1,2] and aeronautics [3,4]. Titanium can be oxidized into titania to gain colourful

appearance which finds its applications in building decoration [5–11] and jewellery [12–14]. Furthermore, Since Japanese scientists Fujishima and Honda first used semiconductor $TiO_2$ to decompose water into hydrogen and oxygen in 1972 [15], the photocatalytic material technology using semiconductor as photocatalyst has become one of hot research directions in the field of clean energy. As we know, great effort has been devoted by researchers around the world to the photoelectric catalysis of semiconductor $TiO_2$ under simulated sunlight, which is widely used in photocatalyst, sensors, biomedicine and other fields [16–26]. The main challenge is still the low photoelectrical conversion efficiency and stability of photocatalytic materials [27–29].

Titanium oxide film can be compulsorily grown by thermal oxidation, laser oxidation and anodic oxidation. Among these methods, anodic oxidation is the most popular method because of its effectiveness and convenience. The oxide materials of titanium sheets prepared by the anodic oxidation method have properties of good photocatalytic activity, high chemical stability, excellent biocompatibility, low cost and non-toxicity, etc. [30–36]. However, there are no reports focusing on both decorative and photoelectrical properties of titanium oxide film obtained by anodization.

In this work, titanium oxide films were prepared by anodic oxidation method on the surfaces of titanium sheets in sodium gluconate electrolyte with the advantage of environmental protection and safety, while different colours were obtained by adjusting duration and voltage in the anodization process. Morphology, crystal structure and photoelectrical performance of anodized titanium sheets were studied. Results show that anodized titanium sheets have excellent photoelectrical performance under both simulated sunlight and visible light, which means absorbed light energy can be converted into electrical energy efficiently by the titanium oxide films on the titanium sheets. From the perspective of energy utilization, the application of anodized titanium sheet as photoeletrical catalyst is supposed to be explored in further work.

# 2. Experimental

## 2.1. Material and methods

In this work, rectangular commercial pure titanium (99.8% purity) sheets with dimensions of $25 \times 10 \times 1$ mm were used as anodes. The cathode was also a titanium sheet. Hydrofluoric acid (HF, 40%) used in the experiment was purchased from Aladdin Biological Technology Co., Ltd (Shanghai, China). Anhydrous sodium carbonate and sodium hydroxide were purchased from Knoc Technology Development Co., Ltd (Tianjin, China). Hydrogen peroxide and sodium gluconate were purchased J&K Chemicals Technology Co., Ltd (Beijing, China). All solutions were prepared with ultra-pure water. All chemicals used in the experiment were analytical reagent (AR) grade.

The surface morphology of anodes was characterized by scanning electron microscopy (FE-SEM, ZeissUltra55, Germany). X-ray diffraction (XRD) patterns were obtained through an UItima IV diffractometer (Japan). Raman spectra were collected on a Micro-Raman spectrometer RM 2000 (Renishaw in Via-reflex, 532 nm excitation laser). Anodization currents were recorded by the universal electric meter (Fluck289, USA). The model of xenon lamp used to measure photocurrent was CEL-S500. Photocurrents were measured by the electrochemical workstation (Chi660E, China), manufactured by Chenhua Instrument Co., Ltd (Shanghai, China). Ultraviolet–visible (UV–vis) spectra were taken with an UV-3900H spectrophotometer (Hitachi, Japan).

## 2.2. Preparation of oxide film on titanium sheets by anodizing

Prior to anodization of the surface, titanium sheets were ground on both sides with silicon carbide paper of 400, 600, 1000, 2000 grit in sequence. Final polishing was done using alumina paste (1 µm size) to produce a smooth surface. To further improve homogeneity, titanium sheets were ultrasonicated for 5 min in a mixture of $10 \, g \, l^{-1}$ NaOH and $25 \, g \, l^{-1}$ $Na_2CO_3$ solution, and were pickled for 20–30 s in a mixture of HF (40%), $H_2O_2$ (30%) and $H_2O$ solution whose volume ratio is $1 : 6 : 3$. Every step was followed by rinsing titanium sheets thoroughly with distilled water. Finally, titanium sheets were blown dry in nitrogen. Anodization was carried out in $20 \, g \, l^{-1}$ sodium gluconate solution. The distance between the two electrodes was greater than 4 cm and the temperature of anodizing was room temperature. Both electrodes were connected to a direct current power source and anodization voltage varied between 20 and 140 V, while current data was recorded during the

anodization process. After anodization, titanium sheets were immersed in 100°C distilled water for 30 min, then rinsed with distilled water and blown dry with nitrogen.

## 2.3. Test of photoelectrical properties

Photoelectrical properties were tested using electrochemical workstation with a solution of 0.05 mol l$^{-1}$ $Na_2SO_4$ as electrolyte. The anodized titanium sheet was used as working electrode, counter electrode was graphite rod, and reference electrode was Ag/AgCl. The area of the working electrode exposed to the light in photoelectrical performance tests was kept at 1 square centimetre (10 × 10 mm). A 500 W xenon lamp was placed in front of the working electrode as simulated sunlight source with wavelength region of 200–900 nm.

# 3. Results and discussion

## 3.1. Anodization current recording

Anodization currents of titanium sheets under different anodization conditions were recorded for 600 s and shown in figure 1. In figure 1a, anodization voltage was kept at 100 V, while anodization durations were 10, 30, 60, 180 and 300 min, respectively. In figure 1b, anodization duration was kept at 10 min, while anodization voltages were 20, 25, 40, 60, 80, 90, 100, 120 and 140 V, respectively. As shown in figure 1, at the beginning of anodization, all currents were very high, while they dropped sharply in just a few seconds, and became stable gradually. In figure 1a, currents became stable after about 180 s. And it can be seen in figure 1b that with higher anodization voltage, current reached a greater stable value. It might be because at the beginning of anodization the electric resistance of titanium sheet was very small, therefore, the anodization current was great. With the continuation of anodization, the thickness of titanium oxide film increases, resulting in the increase of electric resistance of titanium sheet and the decrease of anodization current. However, when the thickness of titanium oxide film reaches a certain value which is decided by anodization voltage, the titanium oxide film stops growing and the anodization current also reaches a stable value. The greater the anodization voltage, the greater the value the anodization current stabilizes at and the later the anodization current reaches a stable value [37].

## 3.2. Morphology characterization

SEM images of surfaces of titanium sheets prepared under different anodization conditions and of a pristine titanium sheet are shown in figure 2. At first, anodization duration was kept at 10 min, and anodization voltages were 25, 80, 100 and 140 V respectively. As shown in figure 2a–d, the surface of titanium sheets is relatively smooth and uniform when the anodization voltage was 25 V (figure 2a). However, 'flower-like' particles which are believed to be titanium oxide grains [38,39] can be seen when the anodization voltage reached 80 V (figure 2b), and the number of titanium oxide grains on the surface increased when the anodization voltage reached 100 V (figure 2c) and 140 V (figure 2d). Then, anodization voltage was kept at 100 V, and the anodization durations were 1 h (figure 2e) and 5 h (figure 2f). It can be seen that there is an obvious increase of titanium oxide grains on the surface with the increase of duration.

It can be seen in figure 1 that the anodization current is very high at the beginning of anodization, and the high current might enable the formation of titanium oxide grains on the surface of titanium. Subsequently, since crystallized titanium oxide grains have better electron transport performance than amorphous titanium oxide particles, these initial crystallized grains would act as ion transport channels [40]. Therefore, with the progress of anodization, titanium oxide crystals would aggregate around these small grains and gradually grow into large grains. If the anodization voltage is high enough, grains would be formed as oxygen evolution or electrical breakdown around the surface of titanium sheet, which plays an important role in the formation of $TiO_2$ grains [41].

## 3.3. Characterization of crystal structure

XRD patterns of titanium sheets anodized under voltages of 20, 25, 40, 60, 80, 100, 120 and 140 V, respectively, while anodization duration was kept at 10 min are shown in figure 3a(a–h). XRD patterns

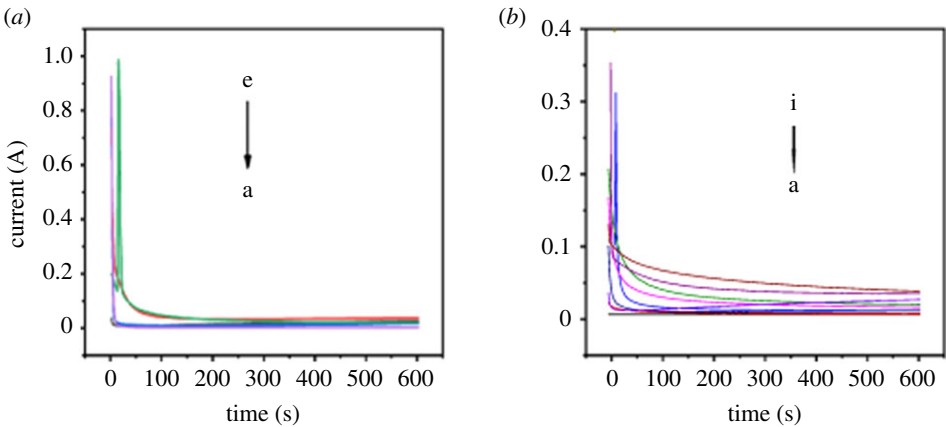

**Figure 1.** Anodization current diagram of titanium sheets under different anodization conditions: anodization voltage was 100 V, and anodization durations were 10, 30, 60, 180 and 300 min, respectively (*a*, a–e); anodization duration was 10 min, and anodization voltages were 20, 25, 40, 60, 80, 90, 100, 120 and 140 V, respectively (*b*, a–i).

**Figure 2.** SEM images of surfaces of titanium sheets: anodized at 25 V, 10 min (*a*), 80 V, 10 min (*b*), 100 V, 10 min (*c*), 140 V, 10 min (*d*), 100 V, 1 h (*e*), 100 V, 5 h (*f*), pristine titanium sheet (*g*).

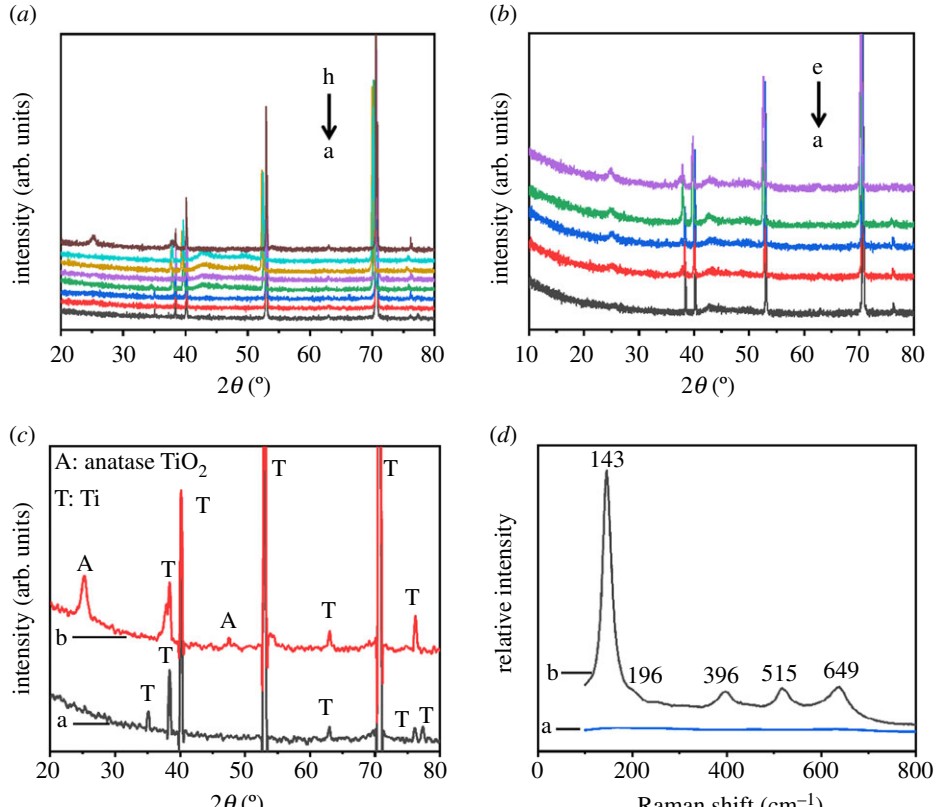

**Figure 3.** XRD patterns of titanium sheets anodized for 10 min under 20, 25, 40, 60, 80, 100, 120 and 140 V, respectively (*a*, a–h). XRD patterns of titanium sheets anodized under 100 V for 10, 30, 60, 180 and 300 min, respectively (*b*, a–e). XRD patterns and Raman spectra of titanium sheets anodized for 10 min under 20 and 140 V, respectively. XRD patterns: 20 V (*c*, a), 140 V (*c*, b). Raman spectra: 20 V (*d*, a), 140 V (*d*, b).

of anodized titanium sheets with durations of 10, 30, 60, 180 and 300 min, respectively, while anodization voltage was kept at 100 V are shown in figure 3*b*(a–e). It can be seen in figure 3*a*(a–h) that when anodization duration was kept at 10 min, only peaks from titanium substrate in the XRD patterns can be found, and peaks of anatase phase of $TiO_2$ ($2\theta = 25°$) did not show up until anodization voltage reached 140 V. In the meantime, it can be seen in figure 3*b*(a–e) that when anodization voltage was kept at 100 V, peaks of anatase phase of $TiO_2$ did not show up until duration reached 30 min, and peaks of anatase phase of $TiO_2$ became intenser with the increase of duration.

To further study the structure of oxide films on the titanium sheets, we compared XRD patterns and Raman spectra of titanium sheets anodized under both 20 and 140 V with duration of 10 min, and the results are shown in figure 3*c*,*d*, respectively. As shown in figure 3(*d*, a), no peak is found when the anodization voltage was 20 V. However, as in figure 3(*d*, b) when the anodization voltage was 140 V, a strong peak at 143 cm$^{-1}$, three moderate peaks at 396, 515 and 649 cm$^{-1}$ and one weak peak at 196 cm$^{-1}$ can be seen. All these peaks match with the Raman spectra of anatase $TiO_2$. Among these peaks, the strong peak at 143 cm$^{-1}$ is caused by bending vibration mode of O–Ti–O, the peak at 396 cm$^{-1}$ is mainly caused by symmetric bending and subordinately caused by anti-scaling of O–Ti–O, the peak at 515 cm$^{-1}$ is mainly caused by antisymmetric bending and subordinately caused by scaling of O–Ti–O, and the peak at 649 cm$^{-1}$ is Raman vibration caused by the symmetric scale of O–Ti–O [42]. The result of Raman spectra is consistent with that of XRD patterns in figure 3*c*, which implies that no crystallized structure would be formed when the anodization voltage is low.

The emerging and increasing intensity of peaks of anatase phase $TiO_2$ implies increasing crystallinity of the obtained titanium oxide films. It is known that increasing crystallinity brings better photoelectrical properties due to improved charge transport properties and decrease of recombination loss. The XRD results are in accordance with the photoelectrical performance shown in §3.5.

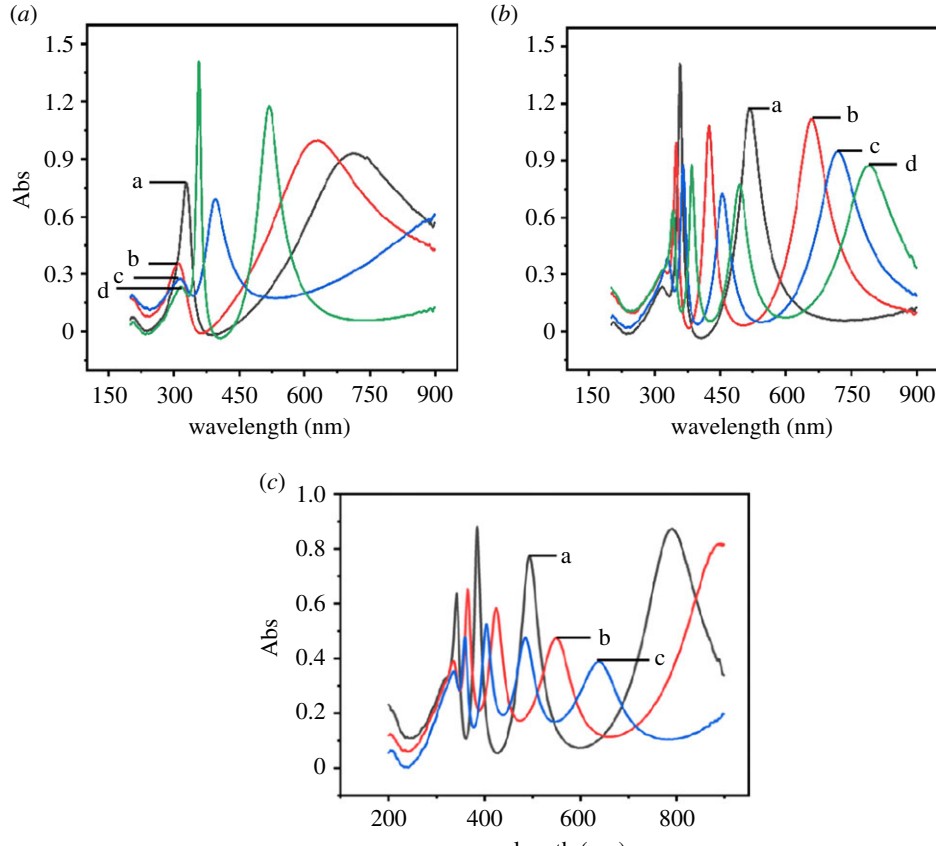

**Figure 4.** UV–vis absorption of titanium sheets anodized for 10 min at different anodization voltages: 20 V (*a*, a), 25 V (*a*, b), 40 V, (*a*, c), 60 V (*a*, d), 60 V (*b*, a), 80 V (*b*, b), 90 V (*b*, c), 100 V (*b*, d), 100 V (*c*, a), 120 V (*c*, b), 140 V (*c*, c).

## 3.4. UV–vis absorption measurement

### 3.4.1. Influence of anodization voltage on UV–vis absorption

UV–vis absorption of titanium sheets anodized for 10 min under different anodization voltages is shown in figure 4. It can be seen in figure 4 that when anodization voltage is in the region of 20–60 V, there is only one absorption peak in the visible light region, viz. 390–780 nm. When anodization voltage is in the region of 80–120 V, two absorption peaks show up. When anodization voltage reaches 140 V, three absorption peaks show up. Generally speaking, the energy band gap of $TiO_2$ nanotubes is 3.2 eV, and its absorption wavelength is about 387 nm, which means that $TiO_2$ nanotubes only absorb light in the ultraviolet region but not in the visible region. $E_g$ values calculated from figure 4 are listed in electronic supplementary material, table S1, which shows that the value of $E_g$ has no obvious relation to anodization voltage when anodization duration was kept at 10 min. The difference between the oxide films we prepared and general $TiO_2$ nanotubes is that the oxide films we prepared can absorb light in the visible light region, and it can be seen in figure 4 that as the anodizing voltage increases, the number of visible light absorption peaks also increases. It might be caused by the enhancement in conductivity brought by introducing $Ti^{3+}$/oxygen vacancy states with the increase of anodization voltage, and significantly decreases the electron transport time and thus leads to an improvement of the photoelectrochemical property. The donor and recipient energy levels were formed between the conductive and valence bands of $TiO_x$, which resulted in the increasing number of absorption peaks of anodized titanium sheets in the visible light region [43,44].

### 3.4.2. Influence of anodization duration on UV–vis absorption

UV–vis absorption of titanium sheets anodized at 100 V with different anodization durations are shown in figure 5. $E_g$ values calculated from figure 5 are listed in electronic supplementary material, table S2, which shows that with the increase of anodization duration, band gap $E_g$ in visible light region also increases, and absorption peak shifts towards long wavelength, which is called the red shift

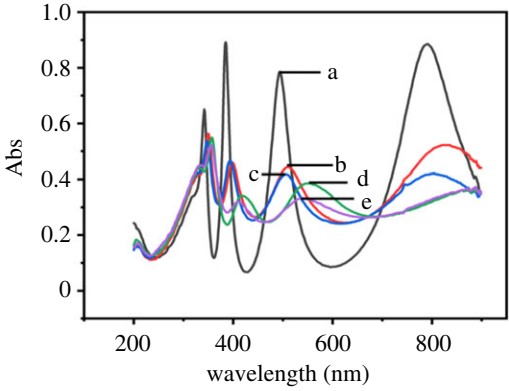

**Figure 5.** UV–vis absorption of titanium sheets anodized at 100 V with different anodization durations: 10 min (a), 30 min (b), 60 min (c), 180 min (d), 300 min (e).

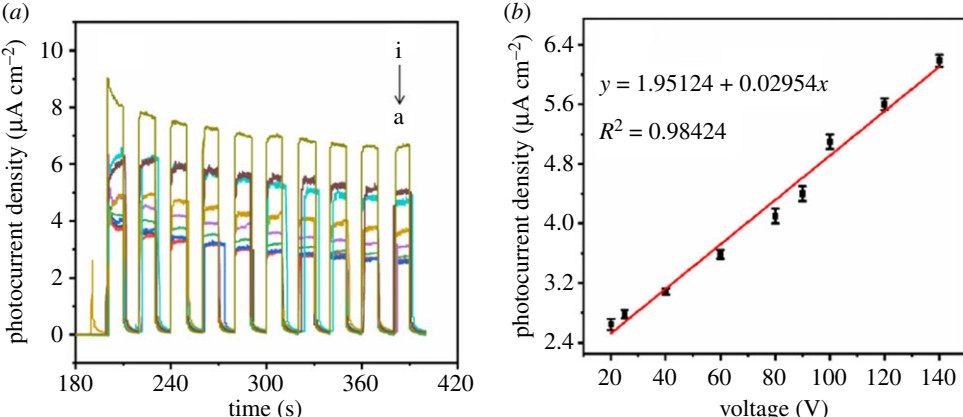

**Figure 6.** Relationship between photocurrent density and anodization voltage: photocurrent density of titanium sheets anodized for 10 min at anodization voltages of 20, 25, 40, 60, 80, 90, 100, 120 and 140 V, respectively, (*a*, a–i) under simulated sunlight and applied voltage of +0.4 V. Linear diagram of photocurrent density and anodization voltage (*b*).

phenomenon. From the equation $E_g$ (eV) = $1240/\lambda_g$, we can figure out that absorption band edge ($\lambda_g$) decreases with the increase of $E_g$, which means energy needed to produce charge transfer transition also decreases. The reason might be that some researchers reported that either doping or oxygen vacancies result in the formation of additional electronic energy level and defect energy level, and the generation of oxygen vacancy plays a fundamental role in decreasing absorption band edge improving photocurrent response of $TiO_x$ under visible light [45–47].

## 3.5. Photoeletrical performance

### 3.5.1. Photoelectrical performance under simulation sunlight

In order to study the influence of anodization voltage and anodization duration on the photoelectrical performance under simulation sunlight, a series of anodized titanium sheets was prepared under different conditions. Firstly, anodization duration was kept at 10 min, and anodization voltages were 20, 25, 40, 60, 80, 90, 100, 120 and 140 V, respectively, and photocurrents of those anodized titanium sheets were tested and converted into photocurrent density in figure 6*a*(a–i). Then, anodization voltage was kept at 100 V, and anodization durations were 10, 30, 60, 180 and 300 min, respectively, and photocurrents of those anodized titanium sheets were tested and converted into photocurrent density in figure 7*a*(a–e). Simulation sunlight was used as light source and the applied voltage was + 0.4 V in the photocurrent testing. We drew an *X–Y* plot in figure 6*b* which used anodization voltage as abscissa and photocurrent density as ordinate, and put the data from figure 6*a* in the plot. The calculation method used in order to obtain photocurrent density can be expressed as

$$\Delta I = I_L - I_D, \tag{3.1}$$

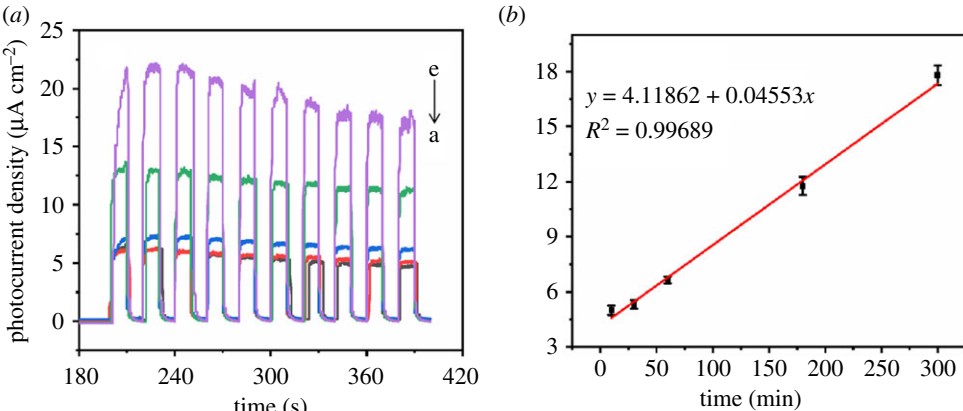

**Figure 7.** Relationship between photocurrent density and anodization duration: photocurrent density of titanium sheets anodized at 100 V for 10, 30, 60, 180 and 300 min, respectively (*a*, a–e) under simulated sunlight and applied voltage of +0.4 V. Linear diagram of photocurrent density and anodization duration (*b*).

where $\Delta I$ is the photocurrent density difference, $I_L$ is the photocurrent density under illumination and $I_D$ is the photocurrent density in the dark [48].

When we connected the photocurrent density dots, a straight line showed up and it was found that photocurrent density increased with the increase of anodization voltage, and the correlation of anodization voltage and photocurrent density matched a unary linear regression equation which can be expressed as

$$I_{pa} = 0.02954x + 1.95124 \quad (R^2 = 0.98424), \tag{3.2}$$

where $I_{pa}$ is the photocurrent density ($\mu A\ cm^{-2}$) and $x$ is the anodization voltage (V).

In the same way, another $X$–$Y$ plot is drawn in figure 7*b* whose abscissa was anodization duration and ordinate was photocurrent density, and the data are from figure 7*a*. When we connected the photocurrent density dots, a straight line showed up and it was found that photocurrent density increased with the increase of anodization duration, and the correlation of anodization duration and photocurrent density also matched a unary linear regression equation which can be expressed as

$$I_{pa} = 0.04553y + 4.11862 \quad (R^2 = 0.99689), \tag{3.3}$$

where $I_{pa}$ is the photocurrent density ($\mu A\ cm^{-2}$) and $y$ is the anodization duration (min).

The reason that photocurrent density shows a good linear correlation with anodization voltage and anodization duration might be: from figure 2, it can be seen that titanium oxide grains increase on titanium sheets with the increase of anodization voltage and anodization duration. Those titanium oxide grains are believed to have better electronic transmission performance than amorphous titanium oxide, and help ions transport and suppress rejoining of electrons and holes, which leads to the increase of photocurrent density [40].

### 3.5.2. Photoelectrical performance under visible light

In order to study the influence of anodization voltage and anodization duration on the photoelectrical performance under visible light, a series of samples under different anodization conditions were prepared. Firstly, anodization duration was kept at 10 min, while anodization voltages were 20, 25, 40, 60, 80, 90, 100, 120 and 140 V, respectively, and photocurrents were tested and converted into photocurrent density in figure 8*a*(a–i). Then, anodization voltage was kept at 100 V, while titanium sheets were anodized for 10, 30, 60, 180 and 300 min, respectively, and photocurrents were tested and converted into photocurrent density in figure 8*b*(a–e). Visible light irradiation of 450 nm filters was used as light source and applied voltage was set at +0.4 V in the tests. It can be seen in figure 8 that the increase of either anodization voltage or anodization duration results in the increase of photocurrent density. It might be because the increase of anodization voltage or anodization duration enhances photon absorption, which prevents $e^-/h^+$ recombination and generates more light-induced electrons [49]. In addition, we think the increase of either anodization voltage or duration brings the increase of oxygen vacancies, thus resulting in the increase of photocurrent density. The reason for the

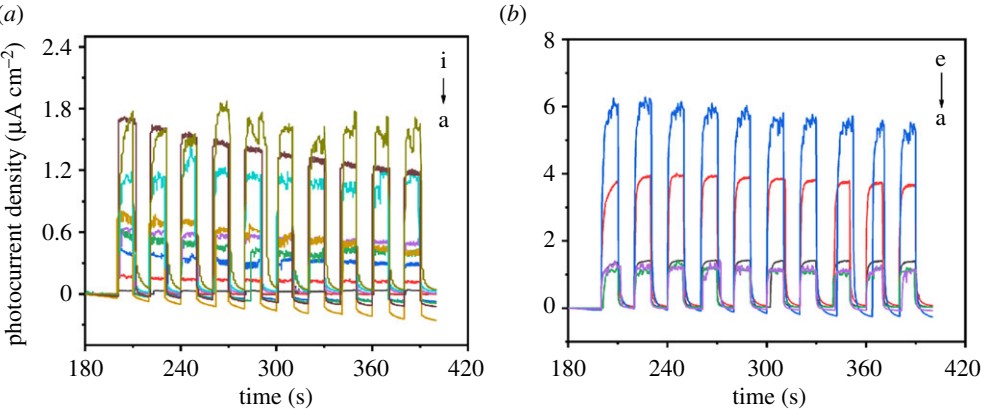

**Figure 8.** Diagram of photocurrent density of titanium sheets anodized for 10 min at voltage of 20, 25, 40, 60, 80, 90, 100, 120 and 140 V, respectively (*a*, a–i); diagram of photocurrent density of titanium sheets anodized at 100 V for 10, 30, 60, 180 and 300 min, respectively (*b*, a–e). All tests were carried out under visible light of 450 nm filters and applied voltage of +0.4 V.

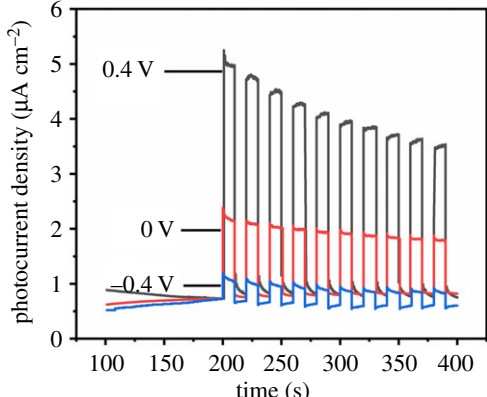

**Figure 9.** Photocurrent density of titanium sheet anodized at 25 V for 10 min under simulated sunlight, at applied voltages of −0.4, 0 and +0.4 V.

formation of oxygen vacancies is probably that $Ti^{4+}$ gains an electron from $O^{2-}$ and transforms into $Ti^{3+}$, while $O^{2-}$ transforms into oxygen and leaves the system. According to figure 5, with the increase of anodization duration of titanium sheets, absorption band edge increases while band gap decreases. Combined with the decrease of band gap, the oxygen vacancy concentration increases, promoting the formation of oxygen vacancy in $TiO_x$ structure and improving the photocurrent response of $TiO_x$ structure under visible light [50].

### 3.5.3. Effect of applied voltage on photoelectrical performance

In order to explore the influence of applied voltages on photoelectrical performance, photocurrents of titanium sheet anodized at 25 V for 10 min were tested under simulated sunlight at applied voltages of −0.4, 0 and +0.4 V. Photocurrents were converted into photocurrent density and are shown in figure 9. It can be seen in figure 9 that photocurrent density decreased slightly across the whole time of recording, which might be caused by: (i) Usually photocurrent increases with the increase of applied voltage. However, when the applied voltage reaches a certain value, no more electrons can be stimulated, resulting in the decreasing tendency of photocurrent, and the tendency might become more obvious with the increase of applied voltage. (ii) At the beginning of being illuminated, anodized titanium sheet releases the most amount of electrons, which results in the highest photocurrent. However, the amount of stimulated electrons is limited and decreases with the increase of illumination duration, which results in the decrease of photocurrent. In addition, it can be seen in figure 9 that photocurrent density increased with the applied voltage shifting positively. It is known

that when titanium oxide is stimulated by light, electron–hole pairs would be photogenerated at the same time. The photogenerated electrons would be forced to the electrode immediately because of the applied voltage, and the holes are transferred to the electrode surface. With the increase of applied voltage, the energy level of the photoelectrode decreases, thus the driving force of electron injection increases, and the recombination probability of photogenerated carriers is reduced. Higher applied voltage results in better photoelectrical performance because of the improving of separation ability of the photogenerated electron–hole pairs and rapid electrochemical reactions on the electrode surface [51].

# 4. Conclusion

A series of multicoloured anodized titanium sheets were prepared under different conditions and characterized. Anodization currents were recorded during anodization and morphology of anodized titanium sheets were observed. When anodization voltage was low, no crystallized titanium oxide film was obtained, while anatase $TiO_2$ was obtained when anodization voltage reached 140 V. It is found that anodized titanium sheets had strong absorption in the visible light region. With the increase of anodization voltage, the number of absorption peaks increased in the visible light region which means increasing absorption. When anodization duration increased, the absorption band edge also increased in the visible light region, which means band gap and energy needed to produce charge transfer transition decreased. In addition, the photoelectrical properties of anodized titanium sheets were studied, and they showed good photoelectrical performance. Under simulated sunlight, photocurrents of anodized titanium sheets were tested and it was found that photocurrent density increased with the increase of either anodization voltage or anodization duration, and can be expressed by linear equations, respectively. The anodized titanium sheets not only can be used as decoration material in jewellery and architecture fields, etc. but also are supposed to be used as photoelectrical catalyst.

Data accessibility. Our data are deposited at Dryad Digital Repository: https://doi.org/10.5061/dryad.d7wm37q04 [52].
Authors' contribution. X.L. and X.C. designed and conducted experiments. X.L. wrote the first draft and X.C. edited the manuscript. L.W. provided full guidance during the research. D.C. made some characterization of samples and corrected some grammar errors of the first draft. All authors read and approved the manuscript.
Competing interests. We declare we have no competing interests.
Funding. This work was financially supported by the Department of education of Guangdong Province in China (grant no. 2019GKTSCX072).

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
