## [Peer Review File · Royal Society Open Science]

Review History

RSOS-201778.R0 (Original submission)

Review form: Reviewer 1

Is the manuscript scientifically sound in its present form?

Yes

Are the interpretations and conclusions justified by the results?

Yes

Is the language acceptable?

Yes

Do you have any ethical concerns with this paper?

No

Have you any concerns about statistical analyses in this paper?

No

Recommendation?

Major revision is needed (please make suggestions in comments)

Comments to the Author(s)

The article deals with the anodization of titanium to produce colored and photoactive titanium oxides. There is a very large number of articles published on the subject; this article presents as novelty the use of sodium gluconate as electrolyte.

On a general basis, the data presented are interesting; yet, the characterization is very basic and in some cases lacks interpretation. To define their material as "suitable photoelectric catalyst"; as the authors do in the conclusions, some photoelectro catalysis tests should be done, not only photocurrent generation. moreover, better correlation between oxide morphology-thickness (never mentioned!)-structure-performance should be given, which is also not presented: the only comment is an observation of increasing photocurrent in crystalline oxides, but this is not surprising.

Moreover, there are some specific issues that need to be addressed:

- English language is mostly ok, but there are some mistakes - pls revise it carefully
- Abstract: it seems like the authors propose the use of anodized titanium as photoactive electrode for the first time. Of course, this is not true. Pls revise the abstract to highlight the actual novelty of this work
- Some terms are used incorrectly. Specifically, in SEM images description, the authors define as "particles" the flower-like structures, but they are rather TiO₂ grains, not particles. Right after that, "the high current might enable the surface of titanium sheet to transform into grains". How can a surface transform into a grain? Maybe the author refer to the fact that the surface is coated with a uniform, transparent oxide and the high current induces the formation of visible grains in it? (which could be ascribed to crystalline grains in an amorphous oxide, which is what also the authors claim, but it is not supported by results in this section, so it should be given as hypothesis). Plus, at page 9 before Fig 6, electrons and holes, not cavities!
- XRD: authors should also give XRD of samples anodized at constant voltage, different time, for completeness. Plus, given the aim of this work and the relevance of crystal structure, the ratio between anatase and rutile phases at the different voltages and/or anodizing times should be computed.
- UV-Vis absorption: data should not be presented in this form, but rather they should be processed to provide the oxides bandgap
- Photocurrent results: why is the relationship linear? is it related to oxide structure or thickness? do the authors have information on oxide thickness? Pus, it is true that there is some activity in visible light, but it is very poor - one order of magnitude lower than that in UV. I would not claim these oxides photoactive in vis light.
- Par 3.5.3 is quite poor - there is no novelty in observing an increase in photocurrent when a current is applied to the working electrode. Moreover, authors claim the photocurrent is stable after 100 s, but I see a decreasing intensity in photocurrent across the whole time of recording (a phenomenon that is not explained by the authors).

Review form: Reviewer 2

Is the manuscript scientifically sound in its present form?

Yes

Are the interpretations and conclusions justified by the results?

Yes

Is the language acceptable?

No

Do you have any ethical concerns with this paper?

No

Have you any concerns about statistical analyses in this paper?

No

Recommendation?

Accept with minor revision (please list in comments)

Comments to the Author(s)

In the reported study, titanium sheets have been subjected to anodization to obtain oxide-based films. Different times and voltages have been used for preparation, then morphological and chemical investigations have been performed to characterize the surfaces. Besides, photoelectrical properties have been investigated.

The paper is technically well written, the context of the research is well clarified and the experimental procedure is properly organized and described. The results are discussed in depth and the conclusions are sound.

In my opinion, minor revision is required before the publication.

- Section 3.2: a SEM image of the pristine surface (i.e., before the anodization could be helpful to highlight the morphological changes/evolution.

- Page 7, lines 8-11: the meaning of the sentence is not clear, please, rephrase. In addition, why do you refer to TiO₂ nanotubes?

- Why did you perform tests under either simulated sunlight or visible light? Please, explain which differences should be expected, especially in term of real applications.

- The manuscript requires a revision of the English style.

Review form: Reviewer 3

Is the manuscript scientifically sound in its present form?

Yes

Are the interpretations and conclusions justified by the results?

Yes

Is the language acceptable?

Yes

Do you have any ethical concerns with this paper?

No

Have you any concerns about statistical analyses in this paper?

No

Recommendation?

Accept with minor revision (please list in comments)

Comments to the Author(s)

In this manuscript, the authors have reported the photoelectrical performance of anodized titanium sheets. A series of multicolored anodized titanium sheets were prepared under different

conditions and characterized. The analysis of experimental data and result is detailed and reliable. However, there are some shirt falls and the work can be improved further for consider in Royal Society Open Science. The following suggestions can be considered.

1. In the introduction part, author should introduce more details the differences between your study and previous study. What are the advantages and innovations of your study?
2. The sentence of “The result of Rama spectra is consistent with that of XRD patterns” spelling error.

Review form: Reviewer 4

Is the manuscript scientifically sound in its present form?

Yes

Are the interpretations and conclusions justified by the results?

Yes

Is the language acceptable?

Yes

Do you have any ethical concerns with this paper?

No

Have you any concerns about statistical analyses in this paper?

No

Recommendation?

Major revision is needed (please make suggestions in comments)

Comments to the Author(s)

Comments: RSOS-201778

This manuscript is about “Study on the photoelectrical performance of anodized titanium sheets”. It seems to be interesting and also explained neatly. The paper may be accepted for publication after clearing/correcting the following suggestions.

1. What is the novelty of this work? The anodized film preparation process is the most common method. Let the authors clarify what this research is like from others and what's so new and have an advantage over other methods. Please make a table summarizing the strengths of this research and other works to confirm the novelty and efficiency of the research.
2. Figure 1B: Why is the current value at a condition of 140V much higher than at any other voltage? Please describe the phenomena that were linked to the results of other studies such as SEM, XRD, and photocurrent.
3. The author should add more study XRD with other applied potential and duration time to clarify the effect of the crystalline structure to the photoelectrocatalytic properties.
4. Due to the UV-Vis absorption of titanium sheets, there are many wavelengths. The author has a choice method and how to calculate the bandgap energy (E_g) from the results? Please explain and summarize the E_g values of Figures 4 and 5.

5. Because the size of the electrode is not equal to 1 square centimeter (25 mm×10 mm×1 mm). Therefore, the photocurrent in y-axis of Fig. 6-9 should be adjusted to the photocurrent density for comparison with other work.

Decision letter (RSOS-201778.R0)

Dear Dr Chen:

Title: Study on the photoelectrical performance of anodized titanium sheets
Manuscript ID: RSOS-201778

Thank you for submitting the above manuscript to Royal Society Open Science. On behalf of the Editors and the Royal Society of Chemistry, I am pleased to inform you that your manuscript will be accepted for publication in Royal Society Open Science subject to minor revision in accordance with the referee suggestions. Please find the reviewers' comments at the end of this email.

The reviewers and handling editors have recommended publication, but also suggest some minor revisions to your manuscript. Therefore, I invite you to respond to the comments and revise your manuscript.

Because the schedule for publication is very tight, it is a condition of publication that you submit the revised version of your manuscript before 19-Dec-2020. Please note that the revision deadline will expire at 00.00am on this date. If you do not think you will be able to meet this date please let me know immediately.

- 1) A text file of the manuscript (tex, txt, rtf, docx or doc), references, tables (including captions) and figure captions. Do not upload a PDF as your "Main Document".
- 2) A separate electronic file of each figure (EPS or print-quality PDF preferred (either format should be produced directly from original creation package), or original software format)
- 3) Included a 100 word media summary of your paper when requested at submission. Please ensure you have entered correct contact details (email, institution and telephone) in your user account

- 4) Included the raw data to support the claims made in your paper. You can either include your data as electronic supplementary material or upload to a repository and include the relevant doi within your manuscript
- 5) All supplementary materials accompanying an accepted article will be treated as in their final form. Note that the Royal Society will neither edit nor typeset supplementary material and it will be hosted as provided. Please ensure that the supplementary material includes the paper details where possible (authors, article title, journal name).

Kind regards,
Dr Laura Smith
Publishing Editor, Journals

On behalf of the Subject Editor Professor Anthony Stace and the Associate Editor Professor Chaohua Cui.

RSC Associate Editor:
Comments to the Author:
(There are no comments.)

RSC Subject Editor:
Comments to the Author:
(There are no comments.)

Reviewer comments to Author:
Reviewer: 1

Comments to the Author(s)
The article deals with the anodization of titanium to produce colored and photoactive titanium oxides. There is a very large number of articles published on the subject; this article presents as novelty the use of sodium gluconate as electrolyte.

On a general basis, the data presented are interesting; yet, the characterization is very basic and in some cases lacks interpretation. To define their material as "suitable photoelectric catalyst"; as the authors do in the conclusions, some photoelectro catalysis tests should be done, not only photocurrent generation. moreover, better correlation between oxide morphology-thickness (never mentioned!)-structure-performance should be given, which is also not presented: the only comment is an observation of increasing photocurrent in crystalline oxides, but this is not surprising.

Moreover, there are some specific issues that need to be addressed:

- English language is mostly ok, but there are some mistakes - pls revise it carefully
- Abstract: it seems like the authors propose the use of anodized titanium as photoactive electrode for the first time. Of course, this is not true. Pls revise the abstract to highlight the actual novelty of this work
- Some terms are used incorrectly. Specifically, in SEM images description, the authors define as "particles" the flower-like structures, but they are rather TiO₂ grains, not particles. Right after that, "the high current might enable the surface of titanium sheet to transform into grains". How can a surface transform into a grain? Maybe the author refer to the fact that the surface is coated with a uniform, transparent oxide and the high current induces the formation of visible grains in it? (which could be ascribed to crystalline grains in an amorphous oxide, which is what also the authors claim, but it is not supported by results in this section, so it should be given as hypothesis). Plus, at page 9 before Fig 6, electrons and holes, not cavities!
- XRD: authors should also give XRD of samples anodized at constant voltage, different time, for completeness. Plus, given the aim of this work and the relevance of crystal structure, the ratio between anatase and rutile phases at the different voltages and/or anodizing times should be computed.
- UV-Vis absorption: data should not be presented in this form, but rather they should be processed to provide the oxides bandgap
- Photocurrent results: why is the relationship linear? is it related to oxide structure or thickness? do the authors have information on oxide thickness? Plus, it is true that there is some activity in visible light, but it is very poor - one order of magnitude lower than that in UV. I would not claim these oxides photoactive in vis light.
- Par 3.5.3 is quite poor - there is no novelty in observing an increase in photocurrent when a current is applied to the working electrode. Moreover, authors claim the photocurrent is stable after 100 s, but I see a decreasing intensity in photocurrent across the whole time of recording (a phenomenon that is not explained by the authors).

Reviewer: 2

Comments to the Author(s)

In the reported study, titanium sheets have been subjected to anodization to obtain oxide-based films. Different times and voltages have been used for preparation, then morphological and chemical investigations have been performed to characterize the surfaces. Besides, photoelectrical properties have been investigated.

The paper is technically well written, the context of the research is well clarified and the experimental procedure is properly organized and described. The results are discussed in depth and the conclusions are sound.

In my opinion, minor revision is required before the publication.

- Section 3.2: a SEM image of the pristine surface (i.e., before the anodization could be helpful to highlight the morphological changes/evolution.
- Page 7, lines 8-11: the meaning of the sentence is not clear, please, rephrase. In addition, why do you refer to TiO₂ nanotubes?
- Why did you perform tests under either simulated sunlight or visible light? Please, explain which differences should be expected, especially in term of real applications.
- The manuscript requires a revision of the English style.

Reviewer: 3

Comments to the Author(s)

In this manuscript, the authors have reported the photoelectrical performance of anodized titanium sheets. A series of multicolored anodized titanium sheets were prepared under different conditions and characterized. The analysis of experimental data and result is detailed and reliable. However, there are some shirt falls and the work can be improved further for consider in Royal Society Open Science. The following suggestions can be considered.

1. In the introduction part, author should introduce more details the differences between your study and previous study. What are the advantages and innovations of your study?
2. The sentence of "The result of Rama spectra is consistent with that of XRD patterns" spelling error.

Reviewer: 4

Comments to the Author(s)

Comments: RSOS-201778

This manuscript is about "Study on the photoelectrical performance of anodized titanium sheets". It seems to be interesting and also explained neatly. The paper may be accepted for publication after clearing/correcting the following suggestions.

1. What is the novelty of this work? The anodized film preparation process is the most common method. Let the authors clarify what this research is like from others and what's so new and have an advantage over other methods. Please make a table summarizing the strengths of this research and other works to confirm the novelty and efficiency of the research.
2. Figure 1B: Why is the current value at a condition of 140V much higher than at any other voltage? Please describe the phenomena that were linked to the results of other studies such as SEM, XRD, and photocurrent.
3. The author should add more study XRD with other applied potential and duration time to clarify the effect of the crystalline structure to the photoelectrocatalytic properties.
4. Due to the UV-Vis absorption of titanium sheets, there are many wavelengths. The author has a choice method and how to calculate the bandgap energy (E_g) from the results? Please explain and summarize the E_g values of Figures 4 and 5.
5. Because the size of the electrode is not equal to 1 square centimeter (25 mm×10 mm×1 mm). Therefore, the photocurrent in y-axis of Fig. 6-9 should be adjusted to the photocurrent density for comparison with other work.

Author's Response to Decision Letter for (RSOS-201778.R0)

See Appendix A.

Decision letter (RSOS-201778.R1)

Dear Dr Chen:

Title: Study on the photoelectrical performance of anodized titanium sheets
Manuscript ID: RSOS-201778.R1

Thank you for submitting the above manuscript to Royal Society Open Science. On behalf of the Editors and the Royal Society of Chemistry, I am pleased to inform you that your manuscript will be accepted for publication in Royal Society Open Science subject to minor revision in accordance with the referee suggestions. Please find the reviewers' comments at the end of this email.

The reviewers and handling editors have recommended publication, but also suggest some minor revisions to your manuscript. Therefore, I invite you to respond to the comments and revise your manuscript.

Because the schedule for publication is very tight, it is a condition of publication that you submit the revised version of your manuscript before 22-Jan-2021. Please note that the revision deadline will expire at 00.00am on this date. If you do not think you will be able to meet this date please let me know immediately.

- 1) A text file of the manuscript (tex, txt, rtf, docx or doc), references, tables (including captions) and figure captions. Do not upload a PDF as your "Main Document".
- 2) A separate electronic file of each figure (EPS or print-quality PDF preferred (either format should be produced directly from original creation package), or original software format)
- 3) Included a 100 word media summary of your paper when requested at submission. Please ensure you have entered correct contact details (email, institution and telephone) in your user account
- 4) Included the raw data to support the claims made in your paper. You can either include your data as electronic supplementary material or upload to a repository and include the relevant doi within your manuscript
- 5) All supplementary materials accompanying an accepted article will be treated as in their final form. Note that the Royal Society will neither edit nor typeset supplementary material and it will

be hosted as provided. Please ensure that the supplementary material includes the paper details where possible (authors, article title, journal name).

Kind regards,
Dr Laura Smith
Publishing Editor, Journals

On behalf of the Subject Editor Professor Anthony Stace and the Associate Editor Professor Chaohua Cui.

RSC Associate Editor

Comments to the Author:

I appreciate the authors' efforts to improve their manuscript. Nevertheless, the authors' point-by-point response to all reviewer feedback should be provided for further evaluation.

Reviewer comments to Author:

Author's Response to Decision Letter for (RSOS-201778.R1)

See Appendix B.

Decision letter (RSOS-201778.R2)

Dear Dr Chen:

Title: Study on the photoelectrical performance of anodized titanium sheets
Manuscript ID: RSOS-201778.R2

It is a pleasure to accept your manuscript in its current form for publication in Royal Society Open Science. The chemistry content of Royal Society Open Science is published in collaboration with the Royal Society of Chemistry.

On behalf of the Subject Editor Professor Anthony Stace and the Associate Editor Professor Chaohua Cui.

RSC Associate Editor
Comments to the Author:
(There are no comments.)

Reviewer(s)' Comments to Author:

Appendix A

Dear Editor,

We are very grateful for the positive evaluation of reviewers to our manuscript (RSOS-201778). We highly value their comments that are helpful for us to revise our manuscript with new contents and corrections. Our detailed responses to the comments are given below. We hope the revised manuscript is acceptable for publication in **Royal Society Open Science**. If you have any further questions, please do not hesitate to get in touch.

Best regards,

Dr Xiangping Chen

Below is our responses to the attached PDF file “Comments RSOS-201778 ” in the email “ **Royal Society Open Science - Decision on Manuscript ID RSOS-201778** ”. Corresponding revisions **are highlighted in red font** in the revised manuscript. Besides, some corrections **are highlighted in yellow background**.

1. **Comment 1:** What is the novelty of this work? The anodized film preparation process is the most common method. Let the authors clarify what this research is like from others and what's so new and have an advantage over other methods. Please make a table summarizing the strengths of this research and other works to confirm the novelty and efficiency of the research.

Response: The advantages of this works are mainly in that: (1) We used Sodium gluconate solution as electrolyte which is environmental-friendly and low-cost. (2) Though the photoelectrical properties of our titanium oxide film prepared by anodization is not as excellent as titanium oxide nanotubes, our anodized titanium sheets are multicolored and suitable to be used as decoration material. (3) No researches on photoelectrical properties of titanium oxide film are reported, and our work investigated the possibility of anodized titanium sheets to be used as both decoration material and photoelectrical catalyst.

Summarized advantages of our work compared to other references are listed in Table 1 below. Corresponding changes are highlighted in red font and located in Pages 1, 2, 14 and 15 in the revised manuscript.

Table 1. Summarized advantages of our work compared to other references

Electrolyte	Environmental friendly	product	Color	Suitable for decoration material	Suitable for photoelectrical catalyst	Corresponding Reference
Glycerol solution and NH ₄ F	No	TiO ₂ nanotubes	Grey	No	Yes	References [23, 24]
Sulfuric acid and hydrofluoric acid	No	TiO ₂ nanotubes	Grey	No	Yes	References [25, 26]
KOH solution	No	TiO ₂ film	multicolored	Yes	No	Reference [35]
Phosphoric acid	No	TiO ₂ film	multicolored	Yes	No	Reference [36]
Sodium gluconate solution	Yes	TiO ₂ film	multicolored	Yes	Yes	This work

2. **Comment 2:** Figure 1B: Why is the current value at a condition of 140V much higher than at any other voltage? Please describe the phenomena that were linked to the results of other studies such as SEM, XRD, and photocurrent.

Response: It is known that the higher anodization voltage, the higher anodization current. However, the current value at 140 V should not be much higher than currents at any other voltages. So we re-did anodization at 140 V, and found that current was not so high as the previous experiment. The original data was replaced in Figure 1B which is located in Page 4.

3. **Comment 3:** The author should add more study XRD with other applied potential and duration time to clarify the effect of the crystalline structure to the photoelectrocatalytic properties.

Response: We add XRD patterns in Fig. 3., such as those of titanium sheets anodized under voltages of 20 V, 25 V, 40 V, 60 V, 80 V, 100 V, 120 V and 140 V respectively while anodization duration was kept at 10 min in Fig. 3 A (a-h), and those of titanium sheets at anodization durations of 10 min, 30 min, 60 min, 180 min, 300 min respectively while anodization voltage was kept at 100 V in Fig. 3 B (a-e). It can be seen from Fig. 3 A (a-h) that when anodization duration was kept at 10 min, there were only peaks from titanium substrate in the XRD patterns, while peaks of anatase phase of TiO₂ ($2\theta = 25^\circ$) showed up when the voltage reached 140 V. In the meantime, it can be seen from Fig. 3 B (a-e) that when anodization voltage was kept at 100 V, peaks of anatase phase of TiO₂ showed up when duration reached 30 min and became intenser with the increase of duration. The emerging and increasing intensity of peaks of anatase phase TiO₂ implied increasing crystallinity of the obtained titanium oxide films. It is known that increasing crystallinity brings better photoelectrical properties due to improved charge transport properties and decrease of recombination losses. The XRD results are in accordance with the photoelectrical performance showed in part 3.5.

Revisions were made in Page 5 and Page 6.

4. **Comment 4:** Due to the UV-Vis absorption of titanium sheets, there are many wavelengths. The author has a choice method and how to calculate the bandgap energy (E_g) from the results? Please explain and summarize the E_g values of Figures 4 and 5.

Response: E_g can be calculated by the equation: E_g (eV) = $1240 / \lambda_g$. However, the determination of λ_g value is uniform and has different methods in different literatures. In this paper, we used the mostly common method to determinate the value of λ_g , namely the the intersection point of the curve tangent to the ultraviolet absorption long wave direction and the baseline. With the values of λ_g in Figures 4 and 5, we calculated E_g values which were listed in Tables 2 and 3 respectively. From Table 2, it can be seen that the value of E_g has no obvious relation to anodization voltage while anodization duration was kept at 10 min. However, from Table 3 it can be seen that the value of E_g decrease with the increase of anodization duration while voltage was kept at 100 V. It means less energy is needed to create the charge transfer transition. The reason might be that some researchers reported that either doping or oxygen vacancies results in the formation of additional electronic energy level and defect energy level, and the generation of oxygen vacancy plays a fundamental role in decreasing absorption band edge and improving photocurrent response of TiO_x under visible light. No revision was made in the manuscript.

Table 2. E_g values calculated from Fig. 4.

Sample	λ_g (nm)	E_g (eV)
20 V, 10 min (Fig. 4A, a)	900+	1.37-
25 V, 10 min (Fig. 4A, b)	900+	1.37-
40 V, 10 min (Fig. 4A, c)	520	2.38
60 V, 10 min (Fig. 4A, d)	640	1.94
80 V, 10 min (Fig. 4B, b)	490, 780	2.53, 1.59
90 V, 10 min (Fig. 4B, c)	525, 900+	2.36, 1.37-
100 V, 10 min (Fig. 4B, d)	590, 900+	2.10, 1.37-
120 V, 10 min (Fig. 4C, b)	490, 670	2.53, 1.85
140 V, 10 min (Fig. 4C, c)	450, 550, 800	2.76, 2.25, 1.55

Table 3. E_g values calculated from Fig. 5.

Sample	λ_g (nm)	E_g (eV)
100 V, 10 min (Fig. 5a)	580	2.14
100 V, 30 min (Fig. 5b)	640	1.94
100 V, 60 min (Fig. 5c)	680	1.82
100 V, 180 min (Fig. 5d)	800	1.55
100 V, 300 min (Fig. 5e)	820	1.51

5. **Comment 5:** Because the size of the electrode is not equal to 1 square centimeter ($25\text{ mm} \times 10\text{ mm} \times 1\text{ mm}$). Therefore, the photocurrent in y-axis of Fig. 6-9 should be adjusted to the photocurrent density for comparison with other work.

Response: The areas of anodized titanium sheets exposed to the light in photoelectrical performance tests were kept at 1 square centimeter ($10\text{ mm} \times 10\text{ mm}$), so the values of obtained photocurrents are equal to their photocurrent densities. Sorry for the missing explanation in experimental part. Y-axes of Fig. 6-9 were modified, and corresponding revisions were made in Page 3 and Pages 9-12.

Appendix B

Dear Editor,

We are very grateful for the positive evaluation of reviewers to our manuscript (RSOS-201778). We highly value their comments that are helpful for us to improve our manuscript with new contents and corrections. Our detailed responses to the comments are given below. We hope the revised manuscript is acceptable for publication in **Royal Society Open Science**. If you have any further questions, please do not hesitate to get in touch.

Best regards,

Dr Xiangping Chen

Below is our responses to the comments of reviewers in the email “ **Royal Society Open Science - Decision on Manuscript ID RSOS-201778** ”. Corresponding revisions are highlighted in red font in the revised manuscript.

Reviewer 1:

Comment 1: English language is mostly ok, but there are some mistakes - pls revise it carefully

Response: We have corrected our manuscript carefully and corresponding corrections are highlighted in red font in the revised manuscript.

Comment 2: Abstract: it seems like the authors propose the use of anodized titanium as photoactive electrode for the first time. Of course, this is not true. Pls revise the abstract to highlight the actual novelty of this work

Response: Thanks for the reviewer's suggestion. Corresponding revisions are highlighted in red font in Abstract part.

Comment 3: Some terms are used incorrectly. Specifically, in SEM images description, the authors define as "particles" the flower-like structures, but they are rather TiO₂ grains, not particles. Right after that, "the high current might enable the surface of titanium sheet to transform into grains". How can a surface transform into a grain? Maybe the author refer to the fact that the surface is coated with a uniform, transparent oxide and the high current induces the formation of visible grains in it? (which could be ascribed to crystalline grains in an amorphous oxide, which is what also the authors claim, but it is not supported by results in this section, so it should be given as hypothesis). Plus, at page 9 before Fig 6, electrons and holes, not cavities!

Response: Thanks for the reviewer's suggestion.

"titanium oxide particles" is revised to "titanium oxide grains". Corresponding revisions are highlighted in red font in part 3.2.

"the high current might enable the surface of titanium sheet to transform into grains" is revised to "the high current might enable the formation of titanium oxide grains on the surface of titanium." Corresponding revisions are highlighted in red font in part 3.2.

"cavities" is revised to "holes". Corresponding revisions are highlighted in red font in part 3.5.1.

Comment 4: XRD: authors should also give XRD of samples anodized at constant voltage, different time, for completeness. Plus, given the aim of this work and the relevance of crystal structure, the ratio between anatase and rutile phases at the different voltages and/or anodizing times should be computed.

Response: Thanks for the reviewer's suggestion. We add XRD patterns in Fig. 3, such as those of titanium sheets anodized under voltages of 20 V, 25 V, 40 V, 60 V, 80 V, 100 V, 120 V and 140 V respectively while anodization duration was kept at 10 min in Fig. 3 A (a-h). XRD patterns of anodized titanium sheets with durations of 10 min, 30 min, 60 min, 180 min, 300 min respectively while anodization voltage was kept at 100 V were shown in Fig. 3B (a-e). It can be seen from Fig. 3A (a-h) that when anodization duration was kept at 10 min, there were only peaks from titanium substrate in the XRD patterns can be found, and peaks of anatase phase of TiO₂ ($2\theta = 25^\circ$) did not show up until anodization voltage reached 140V. In the meantime, it can be seen from Fig. 3B (a-e) that when anodization voltage was kept at 100 V, peaks of anatase phase of TiO₂ did not show up until duration reached 30 min, and peaks of anatase phase of TiO₂ became intenser with the increase of duration. The emerging and increasing intensity of peaks of anatase phase TiO₂ implied increasing crystallinity of the obtained titanium oxide films. It is known that increasing crystallinity brings better photoelectrical properties due to improved charge transport properties and decrease of recombination losses. The XRD results are in accordance with the photoelectrical performance showed in part 3.5. Corresponding revisions are highlighted in red font in part 3.3.

As to the ratio between anatase and rutile phases, there is no rutile phase in the XRD patterns for our anodized titanium sheets, thus the ratio is not computed.

Comment 5: UV-Vis absorption: data should not be presented in this form, but rather they should be processed to provide the oxides bandgap.

Response: Thanks for the reviewer's suggestion. E_g can be calculated by the equation: $E_g \text{ (eV)} = 1240 / \lambda_g$. However, the method to determine λ_g value is not uniform in different literatures. In this paper, we used the mostly common method to determine the value of λ_g , namely the intersection point of the curve tangent to the ultraviolet absorption long wave direction and the baseline. With the values of λ_g in Figure 4 and 5, we calculated E_g values which were listed in Tables 1 and 2 respectively. From Table 1, it can be seen that the value of E_g has no obvious relation to anodization voltage while anodization duration was kept at 10 min. However, from Table 2 it can be seen that the value of E_g decrease with the increase of anodization duration while voltage was kept at 100 V. It means less energy is needed to create the charge transfer transition. The reason might be that some researchers reported that either doping or oxygen vacancies results in the formation of additional electronic energy level and defect energy level, and the generation of oxygen vacancy plays a fundamental role in decreasing absorption band edge and improving photocurrent response of TiO_x under visible light. No revisions are made in the manuscript, and Table 1 and 2 are put in our Electronic Supplementary material.

Table 1. E_g values calculated from Fig. 4.

Sample	λ_g (nm)	E_g (eV)
20 V, 10 min (Fig. 4A, a)	900+	1.37-
25 V, 10 min (Fig. 4A, b)	900+	1.37-
40 V, 10 min (Fig. 4A, c)	520	2.38
60 V, 10 min (Fig. 4A, d)	640	1.94
80 V, 10 min (Fig. 4B, b)	490, 780	2.53, 1.59
90 V, 10 min (Fig. 4B, c)	525, 900+	2.36, 1.37-
100 V, 10 min (Fig. 4B, d)	590, 900+	2.10, 1.37-
120 V, 10 min (Fig. 4C, b)	490, 670	2.53, 1.85
140 V, 10 min (Fig. 4C, c)	450, 550, 800	2.76, 2.25, 1.55

Table 2. E_g values calculated from Fig. 5.

Sample	λ_g (nm)	E_g (eV)
100 V, 10 min (Fig. 5a)	580	2.14
100 V, 30 min (Fig. 5b)	640	1.94
100 V, 60 min (Fig. 5c)	680	1.82
100 V, 180 min (Fig. 5d)	800	1.55
100 V, 300 min (Fig. 5e)	820	1.51

Comment 6: Photocurrent results: why is the relationship linear? is it related to oxide structure or thickness? do the authors have information on oxide thickness? Pus, it is true that there is some activity in visible light, but it is very poor - one order of magnitude lower than that in UV. I would not claim these oxides photoactive in vis light.

Response:

As to the relationship linear, the reason might be that: it can be seen in Fig. 2 that titanium oxide grains increase on titanium sheets with the increase of anodization voltage or anodization duration. Titanium oxide grains are believed to have better electronic transmission performance than amorphous titanium oxide, help ions transport and suppress rejoining of electrons and holes, which leads to the increase of photocurrent.

As to the thickness test of oxide film, the thickness of oxide film obtained by anodization is believed to be 10-100 nm which is not easy to test. It might be the reason that in most references the thickness is calculated instead of being tested. We also failed to find a suitable instrument to test the thickness.

As to the activity in visible light, though it is lower than in UV, our aim is to represent the result that we obtained and the possibility of being used in photoelectrical field for our anodized titanium sheets which is supposed to be explored in further work.

No revisions are made in manuscript.

Comment 7: Par 3.5.3 is quite poor - there is no novelty in observing an increase in photocurrent when a current is applied to the working electrode. Moreover, authors claim the photocurrent is stable after 100 s, but I see a decreasing intensity in photocurrent across the whole time of recording (a phenomenon that is not explained by the authors).

Response: Thanks for the reviewer's suggestion.

It is common in other literatures to observe the change of photocurrent when a voltage is applied to the working electrode, thus we also performed the test.

As to the photocurrent, it did decrease slightly across the whole time of recording, which might be caused by that: (1) Usually photocurrent increases with the increase of applied voltage. However, when the applied voltage reaches a certain value, no more electrons can be stimulated, resulting in the decreasing tendency of photocurrent, and the tendency might become more obvious with the increase of applied voltage. (2) At the beginning of being illuminated, anodized titanium sheet releases the most amount of electrons, which results in the highest photocurrent. However, the amount of stimulated electrons is limited and decreases with the increase of illumination duration, which results in the decrease of photocurrent. Corresponding revisions are highlighted in red font in part 3.5.3.

Reviewer 2:

Comment 1: Section 3.2: a SEM image of the pristine surface (i.e., before the anodization could be helpful to highlight the morphological changes/evolution.

Response: Thanks for the reviewer's suggestion. We add an SEM image of pristine titanium sheet surface in Fig 2G. Corresponding revisions are highlighted in red font in part 3.2.

Comment 2: Page 7, lines 8-11: the meaning of the sentence is not clear, please, rephrase. In addition, why do you refer to TiO₂ nanotubes?

Response: Thanks for the reviewer's suggestion. We think the following paragraph may state some problems that need to be rewritten. The sentence "It might be caused by the central metal titanium ion and the corresponding oxygen vacancy usually form electron states in the band gap with the increase of anodization voltage, then it increases redox electron donor density and improves the charge transfer. " is rephrased to "It might be caused by the enhancement in conductivity brought by introducing Ti³⁺/oxygen vacancy states with the increase of anodization voltage, and significantly decreases the electron transport time and thus leads to an improvement of photoelectrochemical property. " Corresponding

revisions are highlighted in red font in part 3.4.1.

We referred to TiO₂ nanotubes because we found that E_g values of our titanium sheets which are shown in Table 1 and 2 are lower than that of TiO₂ nanotubes, and we think it is the reason why our titanium sheets can absorb light in visible light while TiO₂ nanotubes can't.

Comment 3: Why did you perform tests under either simulated sunlight or visible light? Please, explain which differences should be expected, especially in term of real applications.

Response: Firstly, both our titanium sheets and TiO₂ nanotubes can absorb light under simulated sunlight. However, our titanium sheets can absorb light under visible light while TiO₂ nanotubes can't, which is an advantage of our titanium sheets over TiO₂ nanotubes. Secondly, it might be a theoretical basis of further application research in photoelectrical catalyst field for our anodized titanium sheets, and anodized titanium with excellent photoelectrical property has broader application prospect in both simulated sunlight and visible light regions than in ultraviolet region.

No revisions are made in manuscript.

Comment 4: The manuscript requires a revision of the English style.

Response: Thanks for the reviewer's suggestion. We have made modifications accordingly, which are highlighted in red font in the revised manuscript.

Reviewer 3:

Comment 1: In the introduction part, author should introduce more details the differences between your study and previous study. What are the

advantages and innovations of your study?

Response: Thanks for the reviewer's suggestion. The advantages of this work are mainly in that: (1) We used Sodium gluconate solution as electrolyte which is environmental-friendly and low-cost. (2) Though the photoelectrical properties of our titanium oxide film prepared by anodization is not as excellent as titanium oxide nanotubes, our anodized titanium sheets are multicolored and suitable to be used as decorative material. (3) No researches on photoelectrical properties of titanium oxide film are reported, and our work investigated the possibility of anodized titanium sheets to be used as both decorative material and photoelectrical catalyst.

Summarized advantages of our work compared to other references are listed in Table 3. Corresponding changes are highlighted in red font in Abstract part, Introduction part, and corresponding references are added in References part.

Table 3. Summarized advantages of our work compared to other references

Electrolyte	Environmental friendly	product	Color	Suitable for decorative material	Suitable for photoelectrical catalyst	Corresponding Reference
Glycerol solution and NH ₄ F	No	TiO ₂ nanotubes	Grey	No	Yes	References [23, 24]
Sulfuric acid and hydrofluoric acid	No	TiO ₂ nanotubes	Grey	No	Yes	References [25, 26]
KOH solution	No	TiO ₂ film	multicolored	Yes	No	Reference [35]
Phosphoric acid	No	TiO ₂ film	multicolored	Yes	No	Reference [36]
Sodium gluconate solution	Yes	TiO ₂ film	multicolored	Yes	Yes	This work

Comment 2: The sentence of “The result of Rama spectra is consistent with that of XRD patterns” spelling error.

Response: The sentence of “The result of Rama spectra is consistent with that of XRD patterns” is corrected to “The result of Raman spectra is consistent with that of XRD patterns” . Corresponding revisions are highlighted in red font in part 3.3.

Reviewer 4:

Comment 1: What is the novelty of this work? The anodized film preparation process is the most common method. Let the authors clarify what this research is like from others and what's so new and have an advantage over other methods. Please make a table summarizing the strengths of this research and other works to confirm the novelty and efficiency of the research.

Response: Thanks for the reviewer’ s suggestion. The advantages of this works are mainly in that: (1) We used Sodium gluconate solution as electrolyte which is environmental-friendly and low-cost. (2) Though the photoelectrical properties of our titanium oxide film prepared by anodization is not as excellent as titanium oxide nanotubes, our anodized titanium sheets are multicolored and suitable to be used as decorative material. (3) No researches on photoelectrical properties of titanium oxide film are reported, and our work investigated the possibility of anodized titanium sheets to be used as both decorative material and photoelectrical catalyst.

Summarized advantages of our work compared to other references are listed in Table 3. Corresponding changes are highlighted in red font in Abstract part, Introduction part, and corresponding references are added in References part.

Comment 2: Figure 1B: Why is the current value at a condition of 140V much higher than at any other voltage? Please describe the phenomena that were linked to the results of other studies such as SEM, XRD, and photocurrent.

Response: It is known that the higher anodization voltage, the higher anodization current. However, the current value at 140 V should not be much higher than currents at any other voltages. So we re-did anodization at 140 V, and found that current was not so high as the previous

experiment. The original data was replaced in Figure 1B which is located in part 3.1.

Comment 3: The author should add more study XRD with other applied potential and duration time to clarify the effect of the crystalline structure to the photoelectrocatalytic properties.

Response: We add XRD patterns in Fig. 3., such as those of titanium sheets anodized under voltages of 20 V, 25 V, 40 V, 60 V, 80 V, 100 V, 120 V and 140 V respectively while anodization durations were all kept at 10 min in Fig. 3A (a-h), and those of titanium sheets anodized with durations of 10 min, 30 min, 60 min, 180 min, 300 min respectively while anodization voltage were all kept at 100 V in Fig. 3B (a-e). It can be seen from Fig. 3A (a-h) that when anodization duration was kept at 10 min, there were only peaks from titanium substrate in the XRD patterns can be found, and peaks of anatase phase of TiO_2 ($2\theta = 25^\circ$) did not show up until anodization voltage reached 140V. In the meantime, it can be seen from Fig. 3B (a-e) that when anodization voltage was kept at 100 V, peaks of anatase phase of TiO_2 did not showed up until duration reached 30 min, and peaks of anatase phase of TiO_2 became intenser with the increase of duration. The emerging and increasing intensity of peaks of anatase phase TiO_2 implied increasing crystallinity of the obtained titanium oxide films. It is known that increasing crystallinity brings better photoelectrical properties due to improved charge transport properties and decrease of recombination losses. The XRD results are in accordance with the photoelectrical performance showed in part 3.5.

Corresponding revisions are highlighted in red font in part 3.3.

Comment 4: Due to the UV-Vis absorption of titanium sheets, there are many wavelengths. The author has a choice method and how to calculate the bandgap energy (E_g) from the results? Please explain and summarize the E_g values of Figures 4 and 5.

Response: Thanks for the reviewer's suggestion. E_g can be calculated by the equation: $E_g \text{ (eV)} = 1240 / \lambda_g$. However, the determination of λ_g value is uniform and has different methods in different literatures. In this paper, we used the mostly common method to determinate the value of λ_g , namely the the intersection point of the curve tangent to the ultraviolet absorption long wave direction and the baseline. With the values of λ_g in Figures 4 and 5, we calculated E_g values which were listed in Tables 1 and 2 respectively. From Table 1, it can be seen that the value of E_g has no obvious relation to anodization voltage while anodization duration was kept at 10 min. However, from Table 2 it can be seen that the value of E_g decrease with the increase of anodization duration while voltage was kept at 100 V. It means less energy is needed to create the charge transfer transition. The reason might be that some researchers reported that either doping or oxygen vacancies results in the formation of additional electronic energy level and defect energy level, and the generation of oxygen vacancy plays a fundamental role in decreasing absorption band edge and improving photocurrent response of TiO_x under visible light. No revision was made in the manuscript.

Table 1. E_g values calculated from Fig. 4.

Sample	λ_g (nm)	E_g (eV)
20 V, 10 min (Fig. 4A, a)	900+	1.37-
25 V, 10 min (Fig. 4A, b)	900+	1.37-
40 V, 10 min (Fig. 4A, c)	520	2.38
60 V, 10 min (Fig. 4A, d)	640	1.94
80 V, 10 min (Fig. 4B, b)	490, 780	2.53, 1.59
90 V, 10 min (Fig. 4B, c)	525, 900+	2.36, 1.37-
100 V, 10 min (Fig. 4B, d)	590, 900+	2.10, 1.37-
120 V, 10 min (Fig. 4C, b)	490, 670	2.53, 1.85
140 V, 10 min (Fig. 4C, c)	450, 550, 800	2.76, 2.25, 1.55

Table 2. E_g values calculated from Fig. 5.

Sample	λ_g (nm)	E_g (eV)
100 V, 10 min (Fig. 5a)	580	2.14
100 V, 30 min (Fig. 5b)	640	1.94
100 V, 60 min (Fig. 5c)	680	1.82
100 V, 180 min (Fig. 5d)	800	1.55
100 V, 300 min (Fig. 5e)	820	1.51

Comment 5: Because the size of the electrode is not equal to 1 square centimeter (25 mm×10 mm×1 mm). Therefore, the photocurrent in y-axis of Fig. 6-9 should be adjusted to the photocurrent density for comparison with other work.

Response: Thanks for the reviewer's suggestion. The areas of anodized titanium sheets exposed to the light in photoelectrical performance tests were kept at 1 square centimeter ($10\text{ mm} \times 10\text{ mm}$), so the values of obtained photocurrents are equal to their photocurrent densities. Sorry for the missed explanation in Experimental part. Y-axes of Fig. 6-9 were modified, and corresponding revisions were made in part 2.3 and parts 3.5.1-3.5.3.